# Validation of an Automated Scoring Algorithm That Assesses Eye Exploration in a 3-Dimensional Virtual Reality Environment Using Eye-Tracking Sensors

**DOI:** 10.3390/s25113331

**Published:** 2025-05-26

**Authors:** Or Koren, Anais Di Via Ioschpe, Meytal Wilf, Bailasan Dahly, Ramit Ravona-Springer, Meir Plotnik

**Affiliations:** 1Center of Advanced Technologies in Rehabilitation, Sheba Medical Center, Ramat Gan 52621, Israel; 2Gray Faculty of Medical & Health Sciences, Tel Aviv University, Tel Aviv 69978, Israel; 3Department of Physiology and Pharmacology, Tel Aviv University, Tel Aviv 69978, Israel; 4The Joseph Sagol Neuroscience Center, Sheba Medical Center, Ramat Gan 52621, Israel; 5The Memory and Geriatric Psychiatry Unit, Sheba Medical Center, Ramat Gan 52621, Israel; 6Sagol School of Neuroscience, Tel Aviv University, Tel Aviv 69978, Israel

**Keywords:** virtual reality, eye tracking, fixation time, cognition

## Abstract

Eye-tracking studies in virtual reality (VR) deliver insights into behavioral function. The gold standard of evaluating gaze behavior is based on manual scoring, which is labor-intensive. Previously proposed automated eye-tracking algorithms for VR head mount display (HMD) were not validated against manual scoring, or tested in dynamic areas of interest (AOIs). Our study validates the accuracy of an automated scoring algorithm, which determines temporal fixation behavior on static and dynamic AOIs in VR, against subjective human annotation. The interclass-correlation coefficient (ICC) was calculated for the time of first fixation (TOFF) and total fixation duration (TFD), in ten participants, each presented with 36 static and dynamic AOIs. High ICC values (≥0.982; *p* < 0.0001) were obtained when comparing the algorithm-generated TOFF and TFD to the raters’ annotations. In sum, our algorithm is accurate in determining temporal parameters related to gaze behavior when using HMD-based VR. Thus, the significant time required for human scoring among numerous raters can be rendered obsolete with a reliable automated scoring system. The algorithm proposed here was designed to sub-serve a separate study that uses TOFF and TFD to differentiate apathy from depression in those suffering from Alzheimer’s dementia.

## 1. Introduction

Eye tracking as a research method dates back to the late 1800s, when it was used to trace eye movements for reading studies without the use of a measuring device [1]. It produced insightful results on eye movement behavior while reading solely through observations. Research conducted by L. E. Juval in 1879 concluded that reading is not a linear process, that the eyes do not travel continuously over each line during the passage, but rather in brief fast movements (saccades) interspersed with pauses, or momentary stops of the eyes (fixations) on informative regions of interest [2]. In the 1960s, computer algorithms were added to eye-tracking devices in order to detect and track the iris as the geometric center of the eye. This was a significant catalyst in eye-tracking technology because it enabled automated data analysis and faster processing [1]. In addition, it has been established that oculomotor behavior is indicative of higher cognitive function [3].

### 1.1. Eye-Tracking Setups—The Added Value of Their Use in Virtual Reality

Eye movements can be captured by intrusive and non-intrusive eye-tracking devices (also called remote). Intrusive devices, such as a head-mounted device (HMD), necessitate physical contact with the user. They are typically more precise than remote devices, such as screen-based eye trackers (also known as desktop-mounted eye tracking/computer-embedded eye tracking) or commercial-grade cameras that mostly consist of fixed setups [4]. For instance, in remote eye-tracking studies, participants are required to sit in front of monitors with mounted cameras or desktop-embedded eye-tracking devices [5]. As a consequence, eye movements are constrained to the frame of a computer screen. This raises the issue of the accuracy of 2D eye-tracking models, as one study suggests that looking at non-areas of interest in a 2D screen-based paradigm may be overestimated [6]. Devices that limit or do not account for head movement can be too restrictive when investigating how people interact in real-world settings, making an HMD more favorable [4,7]. Furthermore, the spatial interactivity of a 3D VR environment is thought to elicit an emotional state similar to that found in the real world, which is associated with, e.g., memory performance [6]. As will be described below, the current study is part of a study that seeks to identify emotional reactivity in response to real-life content presented in VR setups (see herein and in the Section 2). For accurate interpretation, precise knowledge of gazing behavior is needed. The current study is motivated by the need to accomplish this automatically and accurately.

### 1.2. Eye-Tracking Metrics

Saccades are characterized as rapid eye movements with amplitude and direction that aim to reposition the eyes from one target to another following a fixation [8]. They are used in research to better understand executive function and the ability to switch focus [8,9]. These eye movements can include nystagmus, drifts, and microsaccades (small movements that are often regarded as noise), which can limit the interpretation of eye-tracking movements [8]. Fixations are considered to be a voluntary manifestation of attention caused by new information processing [8]. In contrast to fixations, visual information is not processed during saccades; thus, saccadic metrics are not as relevant in the context of perception and visual attention as fixation metrics are [10]. Therefore, in the present study, we focused on the atomization of the identification of eyes’ fixation parameters.

Fixation studies frequently examine the ocular spatial position with relation to the area of interest (AOI) [10,11,12]. The experimental paradigms of AOI involve presenting target objects in the participant’s visual field and measuring their on-target fixation behavior [13,14]. This widely applied fixation-based metric allows for calculations of several parameters, for example, total number of fixations, number of fixations on each AOI, total fixation duration, and time to first fixation on target [8]. One of the challenges of fixation analysis is the necessity to annotate fixation durations [15,16]. This is most commonly carried out via manual scoring, which consists of a frame-by-frame analysis of each participant’s video output (the virtual simulation with a superimposed moving gaze point). The manual process requires a human encoder, which is not only a time-consuming process but also allows for unchecked analysis to be processed, potentially leading to measurement error [5,17,18]. Although there has been tremendous progress in the research and application of eye-tracking studies, current automated scoring techniques are not widespread and lack consensus on how to approach and implement fixation identification algorithms on static and moving AOIs [15,19]. To our knowledge, none of the proposed automated methods that use AOI have been evaluated against human observers (i.e., the ‘gold standard’) [11,13,14].

### 1.3. Related Work

The topic of eye tracking in VR has been thoroughly reviewed in medical and scientific literature [19,20]. Fixation identification is a critical aspect of eye-movement data analysis that has important implications for the study of human behavior. As a result, some automated methods were proposed and used for this purpose (for review: [16,19]). However, the majority of the studies found in the literature either used manual encoding or did not focus on the validation of automatic identification algorithms in comparison with the performance of subjective observers (e.g., [4,21,22]). For studies investigating eye tracking in people suffering from intellectual or motor disabilities, their methodology lacked comparison to healthy controls [23,24].

Komogortsev et al. [25] present a set of qualitative and quantitative scores to assess the performance of an eye movement classification algorithm, assuming that a classification algorithm classifies eye position trace into fixation and saccades.

Okamoto et al. [5] developed gaze position-based evaluation metrics for the analysis of eye-gaze patterns in people with severe motor dysfunction. They used the Tobii Pro Spectrum eye-tracking device that was set up on a monitor in a vertical position, 60–65 cm away from the subject’s face, and calculated two metrics that represent to what extent people can chase an object on a monitor.

Chiquet et al. [6] also investigated eye movements in VR using an HMD and integrated eye-tracking hardware. They validated their measurements using a method called “manipulation check”. The authors calculated fixation durations in corresponding AOIs in comparison to non-corresponding AOIs during visual stimuli. It was computed that fixation durations in corresponding AOIs were an average of 4.56 s, 0.43 s on non-corresponding AOIs, and 0.73 s on open space, which accounts for the total time of stimulus exposure (6 s).

Porras-Garcia et al. [26] used a VR-based embodiment technique and an eye-tracking assessment to examine gender differences in attentional bias toward specific-weight or non-weight related body parts. They used “Ogama” software (version 5.1) [27], a stand-alone application designed to analyze gaze and mouse movements in screen-based environments, to count the number of fixations and TFD on the AOI.

### 1.4. Rationale and Motivation

In the present study, we introduce a validation of an algorithm that automatically detects the time of first fixation on the presented target object (i.e., on the AOI). This time point is referred to hereafter as time of first fixation (TOFF). The new algorithm also determines the total fixation duration (TFD) on the AOI. TOFF and TFD are the measures of interest since, when used collectively, they can reveal high levels of attentional focus on the displayed visuals [15]. Additional methods have recently been published; we survey those in the Appendix A.

The eye-tracking algorithm proposed here was developed to accomplish the objectives of a separate clinical study (herein, “Apathy study”), which uses eye tracking as a diagnostic tool to distinguish between the concurrent mood disorders of apathy and depression in those suffering from Alzheimer’s dementia (AD) [28]. See more details about the Apathy study in the Methods Section. These kinds of scientific–clinical studies (i.e., finding potential eye-tracking-related behavioral markers for diagnosis of apathy) underscore the need for reliable and validated automated eye-tracking tools. The eye-tracking data used here were gathered within the framework of the Apathy study and were used for the validation of the automatic scoring algorithm (i.e., results on the Apathy study are not presented in this contribution).

The objective of the present study is to present our algorithm and its validation. The results show that our algorithm detects TFD and TOFF effectively, with a high degree of correlation with manual raters. Therefore, we propose that our algorithm can facilitate studies that use HMD-based VR systems as a reliable and effective automated tool for scoring eye fixations in VR.

## 2. Materials and Methods

### 2.1. Participants

Data sets of six women, aged 76 ± 7.4 [mean ± SD], and four men, aged 71.5 ± 6.3, were randomly selected from data set of an experimental research study (i.e., Apathy study; see below) evaluating gaze behavior in individuals with mild-to-moderate Alzheimer’s disease (AD), who were regular patients of the Geriatric Psychiatry and Memory Unit at the Sheba Medical Center, Israel. Inclusion and exclusion criteria can be found in Appendix A. Including control participants, one hundred participants were recruited to the Apathy study. The study was approved by the institutional review board (IRB) for all participants. After receiving an explanation about the study, participants were examined by an independent physician who determined their capacity to provide informed consent. Those found to be capable, and who were willing to participate in the study, signed the informed consent form and underwent study procedures.

### 2.2. Apathy Study—Procedure

As mentioned in the introduction, the eye-tracking algorithm proposed here was designed to serve a separate clinical study that aims to establish an objective methodology to distinguish between apathy and depression in the context of dementia. The current state of the art is based on subjective questionnaires: MOCA—Montreal Cognitive Assessment; LARS—Lille Apathy Rating Scale; FAB—Frontal Assessment Battery; GDS—Geriatric Depression Scale, which leads to false diagnosis and the administration of antidepressants to people suffering from apathy rather than depression. Explanations on questionnaires and evaluations used in this study can be found in the Appendix A. We hypothesized that by monitoring eye-tracking behavior when emotional images (e.g., of grandchildren) were presented, we could identify behavioral patterns of apathetic people versus those who were not apathetic even without verbal communication (persons with apathy presumably fixate less time on these emotional pictures). As a result, because the person is immersed in the VR environment and is not influenced by the reactions of others in the room, VR can be used as an optimal tool to carry out this protocol.

The VR procedure in the “Apathy study” was conducted on 97 participants and was generally well tolerated by most of them. Four participants did not complete the VR examination due to several reasons, including agitation and irritability (*n* = 1) and complaints about the task’s difficulty (*n* = 3). Four participants reported experiencing dizziness and nausea after the experiment. All of the participants (the data of which were used in the present study, *n* = 10) completed the experiment and one of them reported experiencing dizziness after the experiment. In addition, participants were interviewed regarding the virtual reality experience via a short feedback questionnaire. Participants rated their immersive experience on a scale that ranges between 0 and 4, where zero represents “not immersive” and four represents “fully immersive”. In addition, the participant rated their enjoyment during the experiment and the difficulty level between 0 and 4 (score of four being the highest level of enjoyment or difficulty). The participants included in the current paper (*n* = 10) rated their immersive experience as 3 ± 0.9 (mean ± SD), representing medium–high immersive experience, their enjoyment during the experiment as 3.1 ± 0.9, representing medium–high enjoyment, and the difficulty level as 0.4 ± 0.7, representing low difficulty level. (Please refer to Table 1).

### 2.3. Apparatus

Virtual reality system—Data were acquired using virtual reality (VR) technology within an HMD set (HTC-VIVE, HTC, Taipei, Taiwan). The VR environment used in this study was developed in Unity 3D software (version 2019.3.3). The software also enabled continuous recording of the 3D coordinates of participant gaze points with a sampling rate of 50 Hz in both temporal and spatial dimensions.

Eye-tracking system—Participant gaze was obtained within the HMD using an eye-tracking device (Pupil Player, Pupil-Labs, Germany) with a sampling rate of 200 Hz, a spatial accuracy of 1°, and a gaze precision of 0.08°. The eye-tracking device records participant’s eyes with two sensors (infrared cameras) and two corresponding infrared illuminators. Head movements are monitored by an accelerometer and gyroscope installed in the HMD. The systems provide output of the gaze position coordinates as well as a video that overlays fixation points on the VR scenery. The plug in of the Pupil Player to Unity 3D (“Hmd-Eyes v1.4”) unifies between the coordinate system of the gaze with the virtual environment.

### 2.4. Visual Stimuli

Before the commencement of the experiment, each participant was given an opportunity to acclimate to the virtual reality environment. This involved presenting a “practice” scene in which participants were instructed to scan the VR surroundings in all directions to locate and identify objects within it. This was carried out to ensure that participants understood how to orient in the environment and were able to identify the VR objects.

During the trials, each participant was exposed to a total of 36 visual stimuli presented on a VR environment. The 36 stimuli were presented over 9 trials (each containing 4 visual stimuli) lasting 200–220 sec each, with a 30 sec break between the trials. The participants were instructed to remain seated quietly, and they were allowed to look at any direction they desired. The whole experiment was conducted for about 40 min.

During the breaks between experimental sessions and throughout the experiment, participants were asked about their experience regarding the virtual reality environment and their level of fatigue. They were informed that they could take temporary breaks in the middle of the experiment or whenever they felt the need for rest. During these breaks, we removed the VR headset, allowing them to rest for several minutes until they felt ready to continue with the experiment. None of the participants (the data of which were used in the present study) reported fatigue during the experimental stages.

Eighteen of the 36 stimuli were stationary AOIs displayed on simulated billboards (20° wide and 16° tall) for 10 s, which was the time window used for eye-tracking evaluations. The other 18 stimuli were presented as ‘bus billboards’ with both stationary and moving (dynamic) components. The dynamic AOI’s displayed on buses were introduced by moving from a distant point to a proximal bus stop within the virtual scene (this period lasted 30 sec). Then, for an additional 8 s it remained stationary at the bus stop (the dimensions of the bus billboard while stationary was 10–15° wide and 15–30° tall). Finally, the bus proceeded in the opposite direction from which it had arrived, leaving the visual field distantly within the virtual scene. The time window used for the eye-tracking analysis regarding the dynamic AOI’s began 2 s before the bus stopped and ended after they continued (i.e., 10 sec in total). Thirty seconds separated the disappearance of one stimulus from the VR environment until the appearance of the next one.

It should be noted that the eye-tracking scheme required for these trials are considered dynamic during which the participant is free to move his/her head and body, as opposed to the static scheme where the head is restrained by a conventional harnessing device [29].

### 2.5. Data Collection

The 3D coordinates of the computed gaze position within the given stimuli (i.e., the AOI) were captured using Unity 3D and then imported into MATLAB (version R2024b) for data analysis. The video output showed each participant’s points of fixation (gaze) as a red dot (participant’s pupil) within a wider green circle (participant’s eyeball; Figure 1b) as they examined both the virtual environment and the AOI (see Figure 1a).

Pupil-lab gaze calibration was performed using a 9-point calibration procedure at the beginning of the experiment. To improve accuracy, an additional calibration process was performed: a scene with reference objects appeared, and the participants were asked to gaze at each of them. This was performed prior and immediately after the experiments, and it allowed us to detect if there was a drift in eye-tracking calibration during the period of the experiment, and to apply post hoc corrections as needed. For the present ten data sets, such post hoc corrections were not required. Video file with a demonstration of the VR scene with eye movement can be found in Appendix A.

### 2.6. Manual Scoring of Eye-Tracking Behavior

Manual scoring was conducted by three independent raters (OK, ADVI, and BD) who annotated the eye-tracking video (i.e., frame-by-frame visual score) in accordance with the following criteria:(1)TOFF—the first instance in which the gaze of the participant entered the AOI (millisecond resolution). For example, if a static stimulus is delivered in the VR environment at 90.000 s and the rater detects the participant’s gaze entering the AOI half a second later, TOFF = 90.500 s.(2)TFD—the sum of all fixation durations on the AOI. For example, if the rater detected the participant’s gaze within the boundaries of the AOI three times (2, 1.5, and 3 s), TFD = 6.500 s.

On average (i.e., among raters and across data sets), manual scoring lasted 1.5 h for each 10 min of video footage.

### 2.7. Algorithm and Statistical Analysis

#### 2.7.1. The Algorithm

To automatically calculate TOFF and TFD during eye-tracking tasks in the VR environment, we developed a MATLAB code, referred to hereafter as “the algorithm” that evaluates the temporal and spatial relationships between the continuous gaze position extracted from the eye tracker and the positions of the AOI displayed in the VR scene (i.e., billboards). The algorithm obtains as an input the 3D gaze positions and the stimulus AOI positions traces. The gaze data were down-sampled from 200 Hz to 50 Hz, and preliminary noise reduction including blinks was performed by the plugin of the Pupil Player to Unity 3D software. Visual stimulus presentations with gaze positions that were calculated during loss of eye tracking for more than 100 msec were removed from the analysis (*n* = 8; 2.2%). A one-dimensional median filter with a 10th-order was applied on each time series in order to smooth the signal and filter out tremors with high-frequency movements and very low amplitudes. Then, the 3D traces were trimmed according to the time window of each stimulus. A binary score was given for each gaze sample if it was within the stimulus AOI or not (e.g., positive value if inside or negative value if outside AOI) using a scoring vector; then, the “enter” and “exit” AOI times were calculated by the differences in the scoring vector. Time intervals outside and inside the AOI were removed if they were shorter than 100 milliseconds. Summing all the time intervals inside the AOI yields the TFD and the first enter AOI time was the TOFF, for a specific stimulus. Figure 2 depicts a flow chart of the algorithm. A fully detailed flow chart of the automated scoring algorithm (Appendix A) and a MATLAB demo code can be found in the Appendix A.

#### 2.7.2. Statistical Analysis

To validate the automatic algorithm, interclass correlation coefficient (ICC) estimates and their 95% confident intervals were calculated using SPSS statistical package version 25 (SPSS Inc, Chicago, IL, USA). The ICC analysis was based on single measures, absolute agreement, and two-way random effects model [30]. An ICC value higher than 0.9 was considered as “excellent”, 0.75–0.9 as “good”, 0.5–0.75 as “moderate”, and less than 0.5 as “poor” [31]. For each individual data set, the ICC values for the TOFF and TFD measures were calculated (i.e., 10 × 2 ICC analyses).

## 3. Results

### 3.1. Total Fixation Duration (TFD)

Figure 3 depicts the means of TFD values for each participant, evaluated over 18 stimuli in nine trials. The TFD means achieved by four different raters (i.e., three manual raters and one automatic rater) are presented for stationary and dynamic AOIs separately (i.e., billboards and ‘bus billboards’). It can be clearly seen that TFD values that were automatically annotated by the algorithm have a similar mean and standard error as the ones obtained manually by the human raters. Furthermore, it can be seen that this symmetry between raters exists in both cases when TFD is annotated over stationary and dynamic AOIs.

### 3.2. Interclass Correlation Coefficients (ICC)

Table 2 summarizes the statistical analysis results, indicating that the automated scoring algorithm is highly reliable. The ICC values confirmed the validity of the automated scoring algorithm for detecting TFD (ICC ≥ 0.982) and TOFF (ICC ≥ 0.999).

Further, we examined the Pearson correlation between automatic and manual scoring for each rater separately. Figure 4 depicts TFD values that were automatically annotated versus values obtained manually by each one of the raters. It is clear that the automatic and the manual scoring systems are entirely consistent (r > 0.988, *p* < 0.0001).

## 4. Discussion

The automated scoring algorithm detected TFD and TOFF effectively, as made evident by the high ICC reliability index values (ICC ≥ 0.982 and ICC = 0.99, respectively). These findings reflect a high degree of correlation as well as agreement between measurements. The study did not differentiate between different stimulus types, given that the algorithm considers fixations within the AOI irrespective of the AOI content. This algorithm has shown initial clinically related efficiency in distinguishing apathy among persons with cognitive impairments [28]. It is of importance that the algorithm is effective both for static and dynamic objects in the virtual scene. For studying life-like situations, this advantage is highly significant. Our findings suggest that a reliable automated scoring system can save the considerable time required for manual scoring among numerous raters. Thus, for the VR research and applications arena, we posit that our algorithm can significantly improve studies that use HMD-based VR systems as a reliable and effective automated tool for scoring eye fixations in VR.

### 4.1. Comparison to the Literature

Komogortsev et al. [25] present a set of qualitative and quantitative scores to assess performance of eye movement classification algorithm; however, these suggested scores are not designed to assess AOI-related measurements (i.e., like the scoring algorithm presented in this study). Okamoto et al. [5] developed gaze position-based evaluation metrics for the analysis of eye-gaze patterns in people with severe motor dysfunction. While this non-wearable type of setup allows the experiment to be carried out at home, without imposing a burden on the participants (i.e., of the HMD), the behavior that can be studied with such setup is rather limited as compared to gazing behaviors that are studied in VR settings (e.g., in the present work). For example, it cannot account for head movements associated with environment scanning, which are common in real-world settings.

Chiquet et al. [6] also investigated eye movements (i.e., fixation duration—same as TFD defined above) in VR using an HMD and integrated eye-tracking hardware. This validation method is ‘functional’, but lacks critical information regarding the algorithm’s accuracy through repeated measurements on different stimulus locations and durations, and without measuring correlation with subjective observers as performed in the current study.

Porras-Garcia et al. [26] used a VR-based embodiment technique and an eye-tracking assessment to examine gender differences in attentional bias toward specific-weight or non-weight-related body parts. However, no validation method was used to evaluate the software’s output measures, which can vary depending on the environment and eye-tracking device.

Unlike commercially available eye-tracking software such as Tobii and Pupil Labs, which often lack publicly documented validation against human annotations, our algorithm’s performance was rigorously evaluated and confirmed through direct comparison with three independent human raters, offering a unique perspective on its accuracy.

As a result, without a comparison to manual scores, automatic eye-tracking algorithms are unable to ensure valid fixation locations and durations. In the present study, we used three subjective raters to validate an automated AOI-based eye-tracking algorithm in order to demonstrate successful TFD and TOFF detection.

### 4.2. Challenges of Real-World Eye Tracking

The studies of visual attention that make use of eye tracking mostly utilize fixed setups. For example, the participant is required to sit in front of a monitor, while on the monitor, an eye-tracking device is mounted (e.g., [27]). Also, the use of eye tracking within the HMD mostly involved a static VR environment (e.g., [26]).

On the contrary, the use of wearable mobile gaze trackers is increasingly growing (e.g., [32]), allowing participants to move freely in the real world, removing the previous restrictions that were limiting visual attention studies to 2D surfaces or to virtual 3D predefined objects. In mobile wearable eye trackers, the scene can be recorded using a front-facing camera, and gaze data collected from eye-tracking cameras can be projected onto this video for post hoc analyses [32]. However, the issue of mapping eye movements recorded in a real-world 3D space is not straightforward. Unlike the case with static conditions (i.e., monitors of VR systems), the relative position of the AOI is constantly changing due to the movement of the participant, with no indication of the location of the participant relative to the AOI. Thus, there is a need to address the challenge of aligning to one coordinate system the locations of the AOI, the human participant, and the point of projection of the gaze. One approach to this issue is by mapping gaze collected in 3D onto a 2D reference image, although it will always be limited as a result of incorrect mappings [32]. Another approach suggests moving gaze mapping to 3D virtual models. Pfeiffer et al. [33] propose the visualization of 3D gaze data onto virtual computer-generated models. In 3D scenarios, however, the user will naturally move around, in contrast to the 2D condition where the users remain seated in front of a computer screen, or in a static VR scene with HMD. Thus, gaze mapping in a real-world setting would require more sophisticated methods to make it as convenient to operate as the existing algorithms that were created for static conditions, such as the one presented in the present study.

### 4.3. Limitations

This study can be viewed as a pilot study, and future research with larger patient cohorts will be needed to further confirm these findings and evaluate their clinical utility. The fact that the algorithm was tested in only one cohort (see Methods) limits the generalizability of the findings to other populations. Yet, it is not anticipated that this algorithm, or other automated eye-position-detecting algorithms, will be challenged by a specific cohort, unless this cohort is inherently suffering from ocular motor symptoms (e.g., with pathological nystagmus). However, future research should examine additional populations.

An additional limiting factor is the fact that the automated scoring algorithm was validated using one type of VR scene and without a comparison with other automatic tools. However, it included validation vs. human raters (i.e., ‘gold standard’) using more than 1000 data points. Further, since implanting AOIs in a VR setting is performed in a common manner in all VR environments, we assume that environment-related bias is rare.

### 4.4. Future Directions

We posit that the algorithm can be improved in several ways. We can generate heat maps within the stimuli AOI to identify spots of interest, which can be further analyzed for cognitive condition. In addition, we can extend the time windows for analyzing the eye exploration measures beyond the stimulus period, which can add pre-stimulus and post-stimulus eye movement information.

The TTFF measure can be implemented for real-time applications when dynamic AOI can be adjusted during the VR experiment. However, the TFD measure is calculated only at the end of the stimulus presentation, but it can be adapted by an accumulated measure in real-time applications.

Furthermore, future research should examine the algorithm on additional populations.

In this work, we emphasized the validation process we performed on our algorithm. We posit that the algorithm itself can be combined with other existing platforms, which allows, for example, combining information on head movements to provide more detailed and accurate information (see list of existing platforms in Appendix A).

## Figures and Tables

**Figure 1 sensors-25-03331-f001:**
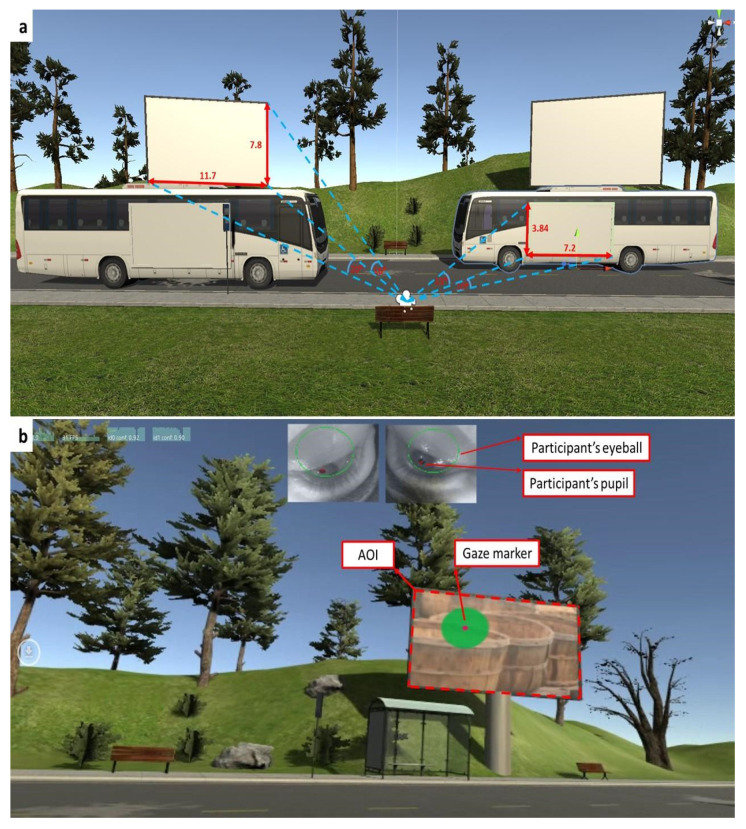
Eye-tracking analysis: (**a**). Area of interest (AOI) dimensions. Both stationary AOIs (i.e., billboards) and dynamic AOIs (bus bill-boards) are presented (see text). (**b**). Gaze tracking. Video outputs include indication of fixation point (red dot within green circle) relative to the area of interest, and optical capturing of the eyes.

**Figure 2 sensors-25-03331-f002:**
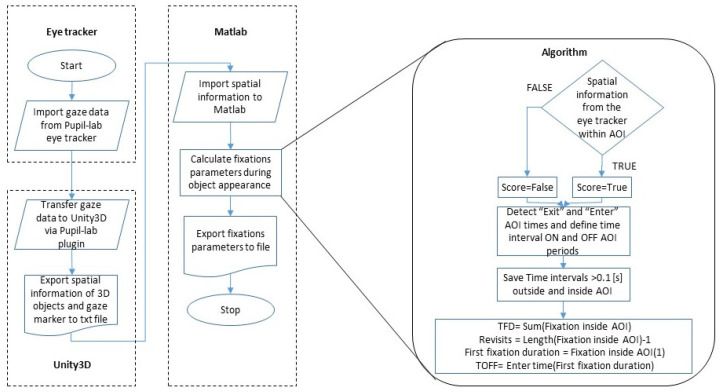
Flow chart of the algorithm. Left—system architecture. Right—detailed scoring methods.

**Figure 3 sensors-25-03331-f003:**
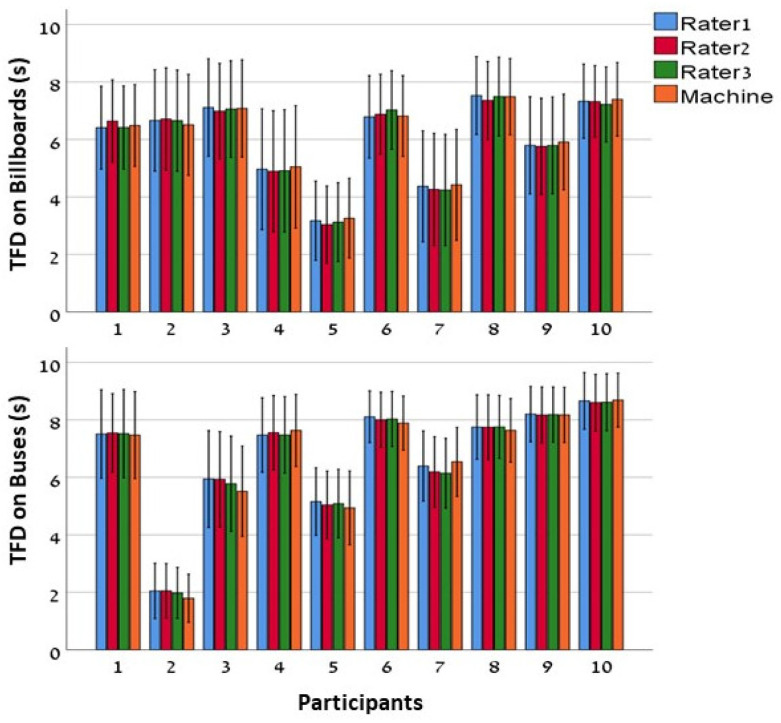
Mean TFD between raters across participants. Group means of TFD are represented by the different colors (see key) for each rater across the participants. The upper and the lower parts represent mean TFD on billboards and ‘bus billboards’, respectively. Error bars—standard error.

**Figure 4 sensors-25-03331-f004:**
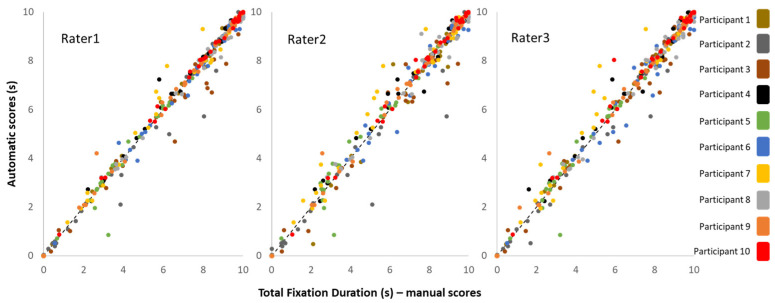
Correlation between Automatic and Manual scoring. Values of TFD (for all ten participants see color key, including all 36 visual stimuli) annotated automatically (ordinate) are plotted against values obtained manually by all three raters (abscissa). It can be observed that values are equally distributed around the unity lines.

**Table 1 sensors-25-03331-t001:** Participants’ clinical and demographic data.

Subject	Gender	Age	MOCA	LARS	FAB	GDS
Participant 1	F	85	14	−12	12	6
Participant 2	M	74	17	−7	9	6
Participant 3	F	84	13	−10	11	0
Participant 4	M	62	19	−9	14	3
Participant 5	M	75	17	−11	14	1
Participant 6	F	68	26	−15	18	1
Participant 7	F	68	29	−7	17	1
Participant 8	F	74	23	−12	15	1
Participant 9	M	75	25	−15	13	0
Participant 10	F	77	26	−7	18	9

MOCA—Montreal Cognitive Assessment; LARS—Lille Apathy Rating Scale; FAB—Frontal Assessment Battery; GDS—Geriatric Depression Scale.

**Table 2 sensors-25-03331-t002:** Interclass Correlation Coefficient results.

Subject	ICC–TFD *	95% Confidence Interval		ICC–TOFF *	95% Confidence Interval	
		Lower Bound	Upper Bound		Lower Bound	Upper Bound
Participant 1	0.987	0.977	0.993	0.999	0.999	0.999
Participant 2	0.985	0.973	0.992	0.999	0.999	0.999
Participant 3	0.984	0.972	0.991	0.999	0.999	0.999
Participant 4	0.992	0.985	0.996	0.999	0.999	0.999
Participant 5	0.985	0.973	0.992	0.999	0.999	0.999
Participant 6	0.983	0.972	0.991	0.999	0.999	0.999
Participant 7	0.982	0.966	0.991	0.999	0.999	0.999
Participant 8	0.991	0.986	0.995	0.999	0.999	0.999
Participant 9	0.996	0.994	0.998	0.999	0.999	0.999
Participant 10	0.993	0.989	0.996	0.999	0.999	0.999

* *p* < 0.0001.

## Data Availability

The raw data supporting the conclusions of this article will be made available by the authors on request.

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
