# Peer review of "Validation of an Automated Scoring Algorithm That Assesses Eye Exploration in a 3-Dimensional Virtual Reality Environment Using Eye-Tracking Sensors"

_sensors, 2025, doi:10.3390/s25113331_

Round 1

Reviewer 1 Report

Comments and Suggestions for Authors

Your paper presents a newly developed system for the automated analysis of ocular movement and fixation in the context of static and dynamic visual stimuli. The automatically generated results are compared with manual measurement runs performed independently by three different examiners. The results demonstrate excellent agreement between the automatically and manually determined results. This excellent agreement, which was achieved in all subjects, is also impressively demonstrated graphically.

The method was calibrated using 100 healthy normal subjects and clinically tested in ten geriatric patients. Due to the small number of subjects, this study can be classified as a pilot study. Of course, the results need to be confirmed further using larger patient cohorts and evaluated for their clinical utility.

Visually induced dysfunction of ocular movement can be impaired in various neurological and psychiatric disorders. The extent of this impairment can correlate with the severity of the respective disease course. This clinical aspect adds value to the paper.

Especially in clinical routine, in contrast to scientific applications, it is extremely beneficial to have automated analysis systems available. This saves personnel and time and also serves to standardize examination procedures. This aspect is also promising – in the sense that the method presented here could potentially be established in the context of routine neurophysiological diagnostics in the future.

Some specific notes and suggestions: 

Table 1: As shown in Table 1, the ten subjects underwent a total of four standardized neurological-geriatric tests. These are identified in the table legend with their names. However, any further explanations of these four scores and classifications of respective score values ​​are missing. Since the readership of the journal "Sensors" is unlikely to be primarily medical, it would be advisable to briefly explain these four test procedures, perhaps not in the full text, but in a separate text box. This would make it easier for readers to classify and categorize the individual values ​​presented. In this context, you could provide the following explanations:

MOCA (Montreal Cognitive Assessment: This dementia test can produce pathological results in numerous neurological and psychiatric disorders, including Alzheimer's disease, vascular dementia, Levy body dementia, frontotemporal dementia, Parkinson's disease, Huntington's disease, stroke, multiple sclerosis, ALS, endogenous depression, schizophrenia, sleep apnea, drug abuse, brain tumors, traumatic brain injury, cerebral involvement in HIV and other systemic infections. Therefore, MOCA is no disease-specific test. It consists of 30 questions, each of which can be awarded one point, resulting in a maximum of 30 points. A total score below 26 points indicates a pathological result in the sense of cognitive impairment.

LARS (Lille Apathy Rating Scale): This test consists of 33 items covering nine different domains. The score is based on a dichotomous scale from -36 to +36. The developers of the test empirically determined a cutoff of -16. Values ​​from -36 to -16 are declared as normal, while values ​​greater than -16 can be regarded as pathological; the higher the numerical value, the more pronounced the apathy. In later applications, a finer gradation was determined and proposed:
-36 to -21: normal, no apathy;
-20 to -16: mild apathy;
-15 to -9: moderate apathy;
> -9: severe apathy.

FAB (Frontal Assessment Battery): This test specifically assesses forms of frontotemporal dementia and can be helpful in differentiating it from Alzheimer's dementia. To assess executive functioning, a total of six categories are used focused on executive functions. A maximum of three points can be awarded per category (normal finding), low graded with 2, 1, or 0 points depending on the severity of dementia. The maximum achievable value (ideal normal score) is 18 points; the theoretically worst score would be zero points. Twelve points are considered the cutoff. Thus, anyone scoring more than twelve points is not considered to have dementia; lower total scores indicate dementia, and the lower the numerical result, the more severe it is.

GDS (Geriatric Depression Scale): This survey instrument consists of 15 individual questions, each of which can be answered with yes or no. If answered affirmatively, this indicates depressive symptoms. The range of measured values ​​thus goes from zero (ideally normal, no depression) to a maximum of 15 (most severe depression). The following gradual classification has been established:
0-5 points: normal findings,
5-10 points: mild to moderate depression,
10-15 points: severe depression.

*****************************************************************

Based on these score descriptions, the following classifications emerge for the ten subjects:

According to the MOCA, three subjects are free from dementia, the others are mildly to moderately demented. According to the LARS, all subjects are affected by apathy, namely seven by moderate apathy and three by severe apathy. According to the FAB, one subject could possibly also be affected by frontotemporal dementia. According to the GDS, only three patients are affected by mild to moderate depression. The remaining subjects are not depressed.

This leads to the conclusion that the LARS scale, at least in these subjects, has by far the highest significance when it comes to discriminating pathological neuropsychiatric deviations from the norm. According to this score, subjects 2, 7 and 10 are most severely affected by apathy; together with subject 4, they each have severe apathy. However, looking at Figure 3, the values of measurements ​​obtained in these subjects do not visibly differ from those of the other subjects. In this context, it could be argued that one should consider clinical aspects in order to correlate the numerical results ​​obtained for each subject with corresponding clinical diagnoses and neuropsychiatric findings. It would also be desirable to present the measurements and values ​​obtained from normal subjects so that these values could be compared with corresponding findings in the geriatric patients. These aspects are not relevant if the focus is on evaluating the measurement method itself in comparison to traditional manual measurements. However, they would be important to derive insights about potential diagnostic benefits in everyday clinical practice.

Regarding Section 2.3: Apparatus: 

A methodological question arises with regard to Section 2.3. You describe that the apparatus is based on four sensors: two infrared cameras and two infrared illuminators. As far as I can see, an illuminator is not a sensor. Only the couple of infrared cameras is fitted with sensors – in normal circumstances: one sensor within one camera. Infrared illuminators, however, are normally not fitted with a sensor. Therefore, in my opinion, it would be more precise to state that the device contained two sensors (infrared cameras) and two corresponding infrared illuminators.

Reviewer 2 Report

Comments and Suggestions for Authors

The reviewer thanks the authors for their commendable work. The article addresses a topical issue: tracking the focus of people’s visual attention. While there are existing solutions to this problem, they do not always work effectively. The authors have proposed a new algorithm designed to recognize the position of an object that captures a person's attention, as well as the duration of that attention. They tested the efficiency of the developed algorithm and compared it with manual processing of video material. The authors suggest that the developed algorithm could be useful in diagnosing certain diseases.

The article is well-structured, with each section clearly addressing its respective topic. The relevance of this research is underscored by the growing body of literature on the subject in recent years. The reviewer did not find any typos or errors in the text or formatting of the article.

The reviewer suggests including (not obligatory):

1. Information on potential improvements to the performance of the proposed algorithm (in the text of the article).
2. A fragment of a video file with eye movement and the corresponding position of the focus cursor (in supplymentary files).

Though this recommendations is not mandatory.

Reviewer 3 Report

Comments and Suggestions for Authors

Some related papers are missing. We can discuss the combination of the following methods with the current system. I think accurate gaze estimation and head pose recognition techniques are crucial for the accuracy of the current system: ADGaze: Anisotropic Gaussian Label Distribution Learning for fine-grained gaze estimation; DADL: Double Asymmetric Distribution Learning for head pose estimation in wisdom museum; GCANet: Geometry cues-aware facial expression recognition based on graph convolutional networks; MFDNet: Collaborative Poses Perception and Matrix Fisher Distribution for Head Pose Estimation; Precise head pose estimation on HPD5A database for attention recognition based on convolutional neural network in human-computer interaction.

Reviewer 4 Report

Comments and Suggestions for Authors

Dear Authors,

I have carefully reviewed your manuscript titled "Validation of an Automated Scoring Algorithm that Assesses Eye Exploration in 3-Dimensional Virtual Reality Environment Using Eye Tracking Sensors" and found it to be a well-structured and significant contribution to the field of eye-tracking research in virtual reality (VR). Your work addresses a critical gap by validating an automated scoring algorithm against manual annotations, demonstrating its high reliability for both static and dynamic areas of interest (AOIs). Below, I provide my overall assessment, followed by specific questions and suggestions for clarification.

Strengths of the Study

  1. Innovation and Practical Impact: The development and validation of an automated algorithm to replace labor-intensive manual scoring is a notable advancement. The high interclass correlation coefficients (ICCs ≥ 0.982) for total fixation duration (TFD) and time of first fixation (TOFF) underscore the algorithm’s reliability.
  2. Clinical Relevance: The application of this tool to differentiate apathy from depression in Alzheimer’s dementia is compelling and highlights its potential for clinical diagnostics.
  3. Methodological Rigor: The inclusion of both static and dynamic AOIs, along with validation against multiple raters, strengthens the robustness of your findings. The use of a head-mounted display (HMD) in a 3D VR environment also enhances ecological validity compared to traditional 2D setups.

Questions and Suggestions for Clarification

  1. Algorithm Generalizability:
    • You mention that the algorithm was tested in a single cohort (older adults with mild to moderate Alzheimer’s disease). Could the algorithm’s performance vary in younger or neurologically healthy populations? Have you tested it in such groups?
    • The study focuses on emotional stimuli (e.g., images of grandchildren). Would the algorithm perform equally well with neutral or non-emotional AOIs?
  2. Technical Details:
    • The manuscript describes noise reduction and smoothing techniques (e.g., median filtering). Were any thresholds or parameters adjusted empirically, or were they fixed across all participants? Could these choices introduce bias in certain cases?
    • How does the algorithm handle edge cases, such as very short fixations (<100 ms) or overlapping AOIs in complex scenes?
  3. Comparison to Existing Tools:
    • You briefly compare your work to prior studies (e.g., Komogortsev et al., Okamoto et al.). However, a more detailed discussion of how your algorithm improves upon or differs from commercially available eye-tracking software (e.g., Tobii, Pupil Labs) would be valuable.
  4. Limitations and Future Directions:
    • The manuscript acknowledges the lack of comparison to other automated tools. Do you plan to address this in future work?
    • Could the algorithm be adapted for real-time applications (e.g., dynamic AOI adjustments during VR experiments)?
  5. Data Availability:
    • The "Data Availability Statement" mentions that raw data will be provided upon request. Would you consider sharing the MATLAB code or a demo version of the algorithm to facilitate reproducibility?

Minor Comments

  • The flow chart in Figure 2 is helpful but could benefit from additional annotations (e.g., specific thresholds or decision points).
  • Some acronyms (e.g., LARS, FAB) are defined only in Table 1. Consider expanding them at first use in the main text for clarity.

Conclusion

Your study makes a compelling case for the utility of automated eye-tracking analysis in VR environments, with strong validation against manual scoring. Addressing the questions above would further enhance the manuscript’s impact and transparency. I commend your team for this rigorous and clinically relevant work and look forward to seeing how this tool evolves in future applications.
